# How Market-Oriented Environmental Regulation Impacts the Bamboo Industry in China

Ting Gao [1], Changming Chen [1] and Zhen Zhu [1,2,*]

1   School of Economics and Management, Zhejiang Agriculture & Forestry University, No. 252 Yijin Street, Hangzhou 311300, China; alice770409@163.com (T.G.); c470777841@gmail.com (C.C.)
2   Zhejiang Province Key Cultivate Think Tank, Research Academy for Rural Revitalization of Zhejiang Province, Zhejiang Agriculture & Forestry University, No. 252 Yijin Street, Hangzhou 311300, China
*   Correspondence: zhuzhen8149278@126.com

**Abstract:** In recent years, stricter environmental rules have affected the entire bamboo industry in China. The increased costs for managing environmental issues hinder the growth of the bamboo forest harvesting and transportation sector in the upstream part of the industry chain. Analyzing how environmental changes affect the entire bamboo industry can give a deeper understanding of the regional transfer within the bamboo industry in China, and it can give more experience to the bamboo industry in developing countries. This study, based on the Pollution Haven Hypothesis and the theory of externalities, collected panel data from 16 provinces in China from 2000 to 2020 and examined the discrepancies in bamboo industry development and the varying intensities of market-oriented environmental regulation (MER). By constructing a fixed effects model and employing econometric methods, this study analyzed the spatiotemporal impact of MER on the output value of the bamboo harvesting and transportation industry (BHTI) and explored whether MER is a crucial factor causing the transfer of the bamboo industry across regions. The findings indicate that there is a noticeable shift of BHTI from the eastern to the central and western regions. Additionally, there is a substantial adverse effect of MER on the BHTI output value, especially in the eastern region. This confirms the transfer of industries between regions, which is a novel contribution of this article. Based on the findings of this study, some recommendations have been given for the response to the environmental regulation for bamboo industries in the future.

**Keywords:** environmental regulation; bamboo harvesting and transportation industry; spatiotemporal variations; industry transfer; China





## 1. Introduction

Bamboo exhibits rapid growth [1], a short rotation period [2], early maturation, and high yield, which are effective in reducing a country's high dependence on foreign countries for wood resources. Bamboo is a traditional and significant forestry resource [3], holding substantial economic and social value in daily life and forestry production; also, it is an important income source for rural villagers around the world [4]. Over the last three decades, it has transitioned from a source of raw materials for basic goods to a basic material for a diverse range of products [5], and there is a trend of using bamboo instead of plastic [6] or wood [7,8]. As one of the biggest developing countries in the world, China, home to over 800 bamboo species [9], boasts more than 6.67 million hectares of bamboo forests, constituting about 66.8% of the total land area [10]. The country has strategically utilized bamboo to foster economic growth. The bamboo industry contributes significantly to the country's GDP, with an annual output value of nearly USD 445,510.4 million [11]. Bamboo's rapid growth and regenerative nature position it as an environmentally sustainable resource. The cultivation and utilization of bamboo contribute to reforestation and mitigate the environmental impact of traditional timber harvesting. The bamboo industry has also become a green industry with great development potential in the new era [12].

However, the Chinese bamboo industry has encountered challenges, particularly noticeable in the southeast coastal areas of Zhejiang Province. The sector experiences periodic declines in output, with the industry's output value dropping from USD 7.4 billion in 2020 to USD 6.98 billion in 2021 [13]. Some areas are even caught in an awkward dilemma, such as the prices of bamboo falling by half with the labor cost of bamboo management rising by half, which caused the profit to be reduced by half in recent years [13]. The residue of bamboo processing will bring some pollution and damage to the surrounding environment. As part of the nationwide efforts to control environmental pollution, the bamboo processing industry has undergone an overall decline [14]. In Suichang County, Zhejiang Province, the number of bamboo processing enterprises declined from 103 in 2010 to 21 in 2020, leading to a gradual reduction in the price of bamboo. This decline adversely affects the bamboo harvesting and transporting industry, posing substantial challenges to the overall development of the bamboo sector.

In order to meet the environmental challenges and promote the construction of ecological civilization and sustainable development, environmental policies and regulations, such as industry wastewater treatment and pollutant discharge fees, have been implemented for pollution-intensive enterprises. Market-oriented environmental regulation (MER) has been paid more attention to due to improved economic incentives to gradually achieve sustainable development in recent years [15]. Instead of setting rules or standards, MER focuses on creating economic mechanisms that encourage businesses to voluntarily adopt environmentally friendly practices, including pollution taxes, carbon trading emission systems, and green certifications. Moreover, MER has become a major tool that encourages companies to seek cost-effective ways to reduce environmental impact [15]. Be that as it may, MER also has adverse effects on the bamboo industry, introducing "environmental barriers" for bamboo processing enterprises and impeding development in certain regions, directly limiting the economic activities of relevant enterprises [16]. For small and medium-sized bamboo processing enterprises, in particular, the relatively high costs of environmental pollution control pose significant challenges, resulting in contraction. The reduced demand for raw materials by secondary processing enterprises in the bamboo industry directly impacts the cultivation and transportation of bamboo forests in the primary sector. Furthermore, the higher labor costs [17] nearly halved the bamboo harvesting price from over USD 5.56 per 100 pounds in the early a 21st century. The decline dampened the enthusiasm of practitioners for forest management, adversely affecting the bamboo industry chain. Given this backdrop, the implementation of MER is expected to result in a regional transfer within the bamboo industry. Drawing on the "pollution haven" hypothesis and the theory of externalities, the eastern region, subject to stringent environmental regulations, is likely to witness a gradual industrial shift towards the western region. This holds substantial practical importance for assessing the differentiated effects of environmental regulation policies in the eastern, central, and western regions of China on the bamboo harvesting and transportation industry (BHTI). Furthermore, it contributes to a deeper understanding of the laws and mechanisms underlying the regional transfer within the bamboo industry, and it is valuable to give more experience to the bamboo industry in developing countries.

Both Chinese and international scholars have studied environmental regulation and the bamboo industry, focusing on two aspects. First, analyze the development of the bamboo industry, evaluate its social and economic benefits, and assess the influence of environmental regulation on its progress. Robust and sustainable growth in the bamboo industry brings economic, social, ecological, and cultural advantages, which, in turn, enhance the overall competitiveness of the industry [18]. The expansion of the bamboo industry contributes to the structural reform of the agricultural supply side, promotes the employment and income of bamboo farmers [19,20], and supports rural revitalization. Moreover, the bamboo industry helps alleviate rural poverty in mountainous areas [21] as it promotes local social and economic progress [18]. Second, scholars have explored the impact of environmental regulation on the bamboo industry. Environmental regulation escalates the environmental protection costs of enterprises [22] and production costs [23]. This, in

turn, forces enterprises to close or transform. On the other hand, some scholars posit that environmental regulation effectively drives the adjustment and upgrade of industrial structures through technological innovation [24]. However, environmental regulation reaching a certain inflection point may affect the industrial structure [25]. Moreover, environmental regulations can stimulate the green innovation of enterprises [26,27], reduce pollution emissions from enterprises [28], and foster high-quality development [29]. Environmental tax is also a regulatory mechanism to incentivize enterprises to adopt environmentally friendly practices [30]. Additionally, regulations may yield unexpected profits for both environmentally responsible and polluting firms [31].

However, few studies have investigated the influence of MER on a regional transfer within the bamboo industry. Some probed into the impact of MER, such as reducing carbon emissions [32] or promoting the green transformation of enterprises [33]. In fact, investigating the spatiotemporal impact of MER on BHTI not only theoretically enriches the research on the influence mechanism of MER on the BHTI, provides a theoretical basis for the industrial transfer of bamboo industry and even other industries, but also further clarifies the influence of MER on the upstream section of the bamboo industry value chain in practice, which is crucial to ensure the safe development of the bamboo industry. Therefore, to fill the research gap, this study looked into the spatiotemporal impact of MER on the BHTI.

This study explores the theoretical and practical implications for the transformation and improvement in the bamboo industry, building upon the externality and pollution haven hypothesis. It involves outlining the conceptual framework of and presenting two hypotheses.

## 2. Conceptual Framework

Environmental regulation is designed to minimize pollution, safeguard the environment, conserve energy, and curtail emissions. It operates as a regulatory force exerted upon individuals or organizations through tangible systems, market incentives, or heightened environmental awareness [34]. In accordance with distinct subject and regulatory mechanisms [35], environmental regulation is generally classified into two categories: mandatory environmental regulation; and MER. The efficacy of mandatory environmental regulation is significantly compromised by the prevalence of "rent-seeking behaviors" during its execution. Consequently, this study focuses on MER.

MER operates via a market mechanism, employing a range of incentives to guide enterprises and individuals in reducing emissions and protecting the environment. The main market-oriented policy tool for pollution control in China is the pollution charge system [36]. The government usually establishes an emission trading system for polluters to buy and sell allowances or permits, including pollutant discharge fees, tradable pollutant discharge permits [37], environmental taxes, pollutant discharge indicators, environmental subsidies, pollutant discharge permits, and ecological compensation. In this study, MER refers to the adoption of a pollutant discharge fee, a successful means to reduce emissions and protect the environment [38].

Bamboo enterprises characterized by high pollution emissions incur tremendous pollution-related costs during their operations, as production and processing inherently cause environmental pollution. Pollutant discharge fees put a financial burden on these enterprises and potentially impede innovation [39]. If an enterprise finds it challenging to meet the financial demands of pollutant discharge fees, it may be unable to sustain operations and, hence, close or transform. This, in turn, directly reduces the demand for primary bamboo production, adversely impacting the growth of the BHTI output value and impeding its overall development.

The expansion of the bamboo industry has given rise to adverse externalities on the environment, primarily from the discharge of pollutants and related issues. From the perspective of environmental economics, where the environment is considered both a limited and communal resource, it is imperative to mitigate the risk of the "tragedy of the

commons" [40]. Therefore, environmental regulatory measures are essential to compel those responsible for environmental degradation to shoulder the costs of environmental governance in order to reduce the negative externalities associated with pollution emissions, which also contributes to the increase in socioeconomic welfare [41].

MER tools are taking the place of mandatory environmental regulation tools, signifying an increasing significance in environmental governance [42] as MER becomes more imperative for overseeing and constraining environmental pollution in the bamboo industry. MER encourages enterprises and individuals to willingly meet the environmental protection requirements through market-driven mechanisms, thereby achieving environmental conservation goals. It mainly consists of the levying of pollutant discharge fees, the provision of subsidies, and the implementation of tradable pollutant discharge permits [37]. Of particular importance in China is the use of pollutant discharge fees as a policy to internalize environmental externalities. However, MER also elevates the costs of pollution control, affecting bamboo processing enterprises, especially smaller ones. In pursuit of profit maximization, these enterprises are compelled to revise their business decisions by lowering production scale. As a result, the demand for bamboo raw materials is reduced, leading to a declined output value of BHIT. On such basis, we proposed the following hypothesis:

**Hypothesis 1 (H1):** *MER has a negative effect on the growth of the BHTI output value.*

As the international community cares more about environmental preservation, high-emission enterprises are facing stricter environmental regulation. The Pollution Haven Hypothesis posits that stringent environmental regulations in a country may prompt the relocation of pollution-intensive industries to countries with less rigorous regulations, affecting the geographical distribution of industrial production and international trade [43]. This is exemplified by the potential restructuring and spatial redistribution of pollution-intensive industries, as they may shift from countries and regions with stringent environmental regulations to those with comparatively lower environmental standards [44]. Eastern China has advanced socioeconomic development and a well-established bamboo industry, so the rapid growth of the industry has brought environmental challenges. Therefore, stringent environmental regulations have been issued to address the challenges. This, however, has put more constraints on the economic activities of the local bamboo industry. So, some polluting enterprises in Eastern China may relocate to central and western regions where pollution treatment costs are less prohibitive. Thus, we proposed the following hypothesis:

**Hypothesis 2 (H2):** *Influenced by MER, the bamboo industry undergoes a regional transfer from the eastern to central and western regions.*

To clarify, a conceptual framework was constructed (Figure 1).

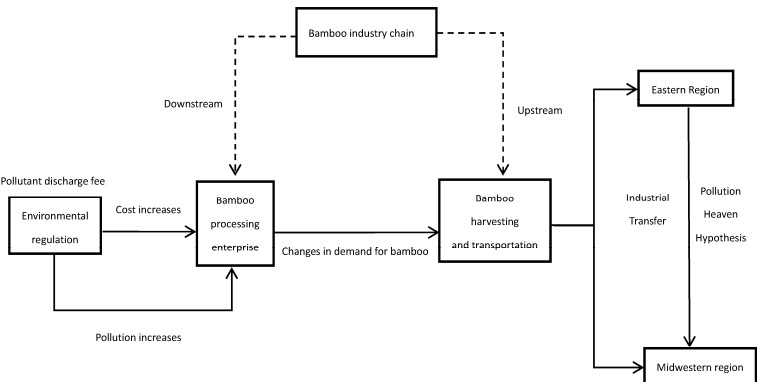

**Figure 1.** Conceptual framework.

### 3. Methods

#### 3.1. Modelling

This study used the F-test to assess the applicability of the mixed effects model, the fixed effects model, and the Hausman test to choose between a fixed effects model and a random effects model. In detail, we tested the individual effects and time effects via Stata15.0. Results of the F-test showed that $p < 0.05$ (individual fixed effect) and $p > 0.05$ (no time-fixed effect), respectively. Results of the Hausman test showed that $p < 0.05$; so, the fixed effects model was used instead of the random effect model. The model is as follows:

$$forest_{i,t} = \alpha_0 + \alpha_1 Regulation_{it} + control_{it}\gamma + \mu_i + \varepsilon_{i,t}$$

where $forest_{i,t}$ refers to the BHTI output value in province $i$ in year t; $Regulation_{it}$ refers to MER intensity in province $i$; $control_{it}$ refers to control variables, including annual temperature, annual precipitation, population density, per capita income, and industrial structure; $\alpha_0$ is a constant term; $\alpha_1$ and $\gamma$ are coefficients; $\varepsilon_{i,t}$ is the disturbance term; $\mu_i$ is an unobserved individual effect and does not change with time.

#### 3.2. Variables

In light of Zsarnóczai, et al. (2019) [45], the development of an industry can be expressed by its output value, so the output value of the bamboo harvesting and transportation industry (*Bopt*) was chosen to represent the development of the BHTI. It is measured by the annual output value produced by BHTI in a province.

The explanatory variable is MER. The traditional mandatory environmental regulation, represented by government intervention, faces numerous challenges, such as the high cost of policy implementation, low efficiency, "rent-seeking behavior", and endogenous law enforcement. These challenges greatly reduce the effectiveness of traditional regulatory approaches. Therefore, there is a growing trend toward the adoption of MER. Feng, et al. (2016) [46] and Zhang, et al. (2023) [47] proposed to use pollution control costs and pollutant discharge as metrics to measure the intensity of MER. Additionally, Li et al. (2020) [48] identified two main methods for constructing an environmental regulation index. The first method involves using a single index, encompassing indicators like environmental policy performance and environmental governance investment. The second method, known as the comprehensive index method, measures regulation intensity based on the overall index of pollution emissions. Due to the data availability, we denoted the MER by **the pollutant discharge fee** (*fee*). That is the annual pollution charge of each province.

Other variables were employed in this study. Annual average temperature (*tem*), annual average precipitation (*pre*), population density (*pop*), per capita income (*inc*), and industrial structure (*stru*) were selected as control variables. Bamboo is usually grown in temperate, tropical, and subtropical zones, affected by temperature and rainfall. So, both annual average temperature and annual average precipitation were selected as control variables. The increase in population density signifies progress in local urbanization, which consumes land resources for industrial, construction, and residential use, leading to a decline in forest resources, including bamboo forests. Population density is measured by the total population per land area in each province. Bamboo forest management needs a certain economic foundation, and environmental regulation is beneficial in increasing farmers' agricultural production income [12]. Therefore, per capita income was chosen as an indicator to reflect the economic development level of families engaged in the bamboo industry. The optimization of the regional industrial structure will promote the integration of rural industries and vitalize the agricultural economy [49]. Therefore, industrial structure optimization will affect BHTI—one of the traditional agriculture and forestry industries.

#### 3.3. Data Collection

It can be found that the bamboo forest area in China reached 7.5627 million hm². China's bamboo forests are distributed across 20 provinces [50]. Due to the availability

of the data and the remaining four provinces having very little distribution of bamboo forests, only 16 provinces were selected for this study (shown in Figure 2). Among these, 8 provinces boast forest areas exceeding 300,000 hm² each. These provinces, namely, Fujian, Jiangxi, Hunan, Zhejiang, Sichuan, Guangdong, Guangxi, and Anhui, collectively cover a total area of 6.785 million hm², accounting for 89.72% of the bamboo forest area in China (NFGA, 2023).

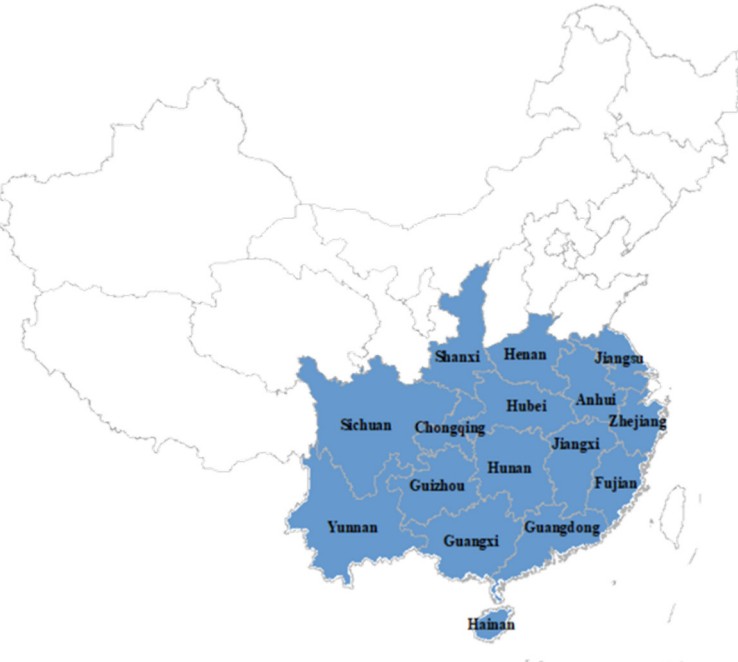

**Figure 2.** The bamboo distribution of 16 provinces in China.

It should be noted that the data of bamboo harvesting and transportation (i.e., *Bopt*) are counted together in China; so in this paper, they are studied as a whole, which all belong to the primary industry in the bamboo industry. Therefore, this paper studies the influence of MER on the bamboo processing enterprises belonging to the secondary industry of the bamboo industry so as to further influence the BHTI of the primary bamboo industry, which can be understood as the influence mechanism among the three subjects.

This study collected panel data from 16 provinces in China from 2000 to 2020, with 2000, 2010, and 2020 as the main sample years. To ensure accuracy and reliability, the data were mainly from yearbooks and relevant documents released by the environmental protection departments, including https://pkulaw.com/, China Meteorological Data Service Centre (an authoritative and unified shared service platform for China Meteorological Administration), *China Statistical Yearbook*, *China Forestry Statistical Yearbook*, National Forest Inventory Report, *China Population and Employment Statistical Yearbook* and statistical yearbooks of provinces.

Given both the socioeconomic development and geographical location, the 16 provinces are categorized into three regions: eastern areas (5 provinces: Jiangsu; Fujian; Zhejiang; Guangdong; and Hainan); central regions (5 provinces: Jiangxi; Anhui; Henan; Hubei; and Hunan); and western regions (6 provinces/cities: Shanxi; Sichuan; Chongqing; Guizhou; Yunnan; and Guangxi)—Heilongjiang, Jilin, Inner Mongolia, and Xinjiang are not included, because bamboo forests are not mainly planted in these three provinces. From the economic statistic data of all 16 case provinces (Table 1), it can be found that the average per capita GDP and net income of farmers are USD 15,520 and 3718 in eastern provinces, respectively, which is much higher than in central and western provinces.

**Table 1.** Economic Status of 16 provinces in eastern, central, and western China in 2022.

| Region | Province/City | The Economic Development Indicators | | |
|---|---|---|---|---|
| | | **Provincial Total GDP (Unit: Billion USD)** | **Per Capita GDP (Unit: USD)** | **Rural Per Capita Net Income (Unit: USD)** |
| Eastern | Zhejiang | 10,803.68 | 16,472.84 | 5222.14 |
| | Jiangsu | 17,081.67 | 20,072.52 | 3960.08 |
| | Fujian | 7383.13 | 17,631.26 | 3473.54 |
| | Guangdong | 17,949.55 | 14,166.43 | 3280.47 |
| | Hainan | 947.84 | 9258.74 | 2657.62 |
| | Mean | 10,833.17 | 15,520.36 | 3718.77 |
| Central | Anhui | 6261.98 | 10,231.99 | 2721.22 |
| | Jiangxi | 4458.90 | 9859.43 | 2771.43 |
| | Henan | 8527.95 | 8633.73 | 2599.22 |
| | Hubei | 7470.01 | 12,797.67 | 2739.93 |
| | Hunan | 6765.96 | 10,231.30 | 2717.25 |
| | Mean | 6696.96 | 10,350.82 | 2709.81 |
| Western | Shanxi | 4555.93 | 11,519.42 | 2183.16 |
| | Sichuan | 7889.13 | 9422.09 | 2595.76 |
| | Chongqing | 4049.40 | 12,603.61 | 2684.77 |
| | Yunnan | 4025.10 | 8579.51 | 2105.66 |
| | Guizhou | 2803.20 | 7273.46 | 1905.45 |
| | Guangxi | 3656.25 | 7251.63 | 2423.42 |
| | Mean | 4496.50 | 9441.62 | 2316.37 |
| Total | - | 114,629.68 | 186,005.63 | 46,041.12 |

Source of data: The Ministry of Science and Technology of China and the National Bureau of Statistics of China.

## 4. Results and Discussion

### 4.1. Descriptive Analysis

According to Table 2, the BHTI output value (the *Bopt* of the province in that year) varied greatly among provinces. The average output value in 16 provinces was USD 156.56 million; the maximum value was USD 13.16 billion in Fujian Province in 2020, and the minimum value was USD 16,700 in Henan Province in 2000. As per Table 2, the MER intensity varied significantly across provinces. Using the pollutant discharge fee (The *fee* of the province in that year) as the proxy variable for MER intensity, from 2000 to 2020, the average pollutant discharge fee across the 16 provinces was USD 70.81 million. The highest fee was charged by Jiangsu in 2019, reaching USD 49.87 million, while the lowest was recorded in Hainan in 2001—USD 1.61 million, manifesting a substantial gap.

**Table 2.** Descriptive statistics of variables.

| Type | Variable | Description (Unit) | Mean | S.d. | Min | Max |
|---|---|---|---|---|---|---|
| Dependent variable | BHTI output value of case province (*Bopt*) | USD ten thousand | 15,656 | 21,566.75 | 1.67 | 131,553.13 |
| Core explanatory variable | Pollutant discharge fee of case province (*fee*) | USD ten thousand | 7080.62 | 6340.09 | 161.19 | 49,871.44 |
| Control variable | Annual average temperature (*tem*) | °C | 17.66 | 2.95 | 11.82 | 25.44 |
| | Annual average precipitation (*pre*) | mL | 1065 | 303.7 | 491.3 | 1961 |
| | Population density (*pop*) | Person/km$^2$ | 346 | 173 | 109 | 771 |
| | Per capita income (*inc*) | USD ten thousand | 2842.17 | 1694.07 | 662.29 | 8712.72 |
| | Industrial standard (*stru*) | the ratio of GDP of the primary industry to provincial GDP (%) | $1.37 \times 10^{-3}$ | $6.60 \times 10^{-4}$ | $3.36 \times 10^{-4}$ | $3.79 \times 10^{-3}$ |

Source of data: *China Forestry Statistics Yearbook*, *China Environmental Statistics Yearbook*, https://pkulaw.com/.

The bamboo market in the eastern region (Zhejiang and Fujian) was relatively developed but still grappled with challenges related to industrial transformation and upgrading.

Conversely, despite the evident advantages of bamboo resources, the central and western regions lagged behind in the development of the bamboo industry [51]. According to the pollution fee data, the well-established bamboo industry in the eastern region faced a larger influence from environmental regulations. Meanwhile, the bamboo industry in the western region was still in its infancy with fewer regulations, fostering a robust development momentum.

According to Table 3, Fujian Province in the eastern region has witnessed significant growth in its bamboo industry, marked by a substantial increase in output value. While Fujian stood out as a major contributor to bamboo industry development, the growth momentum in output value for other eastern provinces was curbed by regulations. Conversely, most provinces in the central and western regions grew rapidly in terms of output value. This can be attributed to the relatively lenient local environmental regulations and policies. To sum up, despite the higher average output value in the eastern region, the disparity among the three regions was gradually diminishing.

**Table 3.** Bamboo Harvesting and Transportation Output of 16 Provinces in 2020, 2010, and 2020.

| Region | Province | Output of Bamboo Harvesting and Transportation (Unit: USD Ten Thousand) | | |
|---|---|---|---|---|
| | | **2000** | **2010** | **2020** |
| East | Zhejiang | 399.22 | 33,659.69 | 40,276.27 |
| | Jiangsu | 52.11 | 795.39 | 3307.73 |
| | Fujian | 619.89 | 49,969.49 | 128,654.03 |
| | Guangdong | 113.67 | 17,019.11 | 43,134.47 |
| | Hainan | 14.17 | 870.31 | 277.64 |
| | Mean | 239.85 | 20,463.26 | 43,130.99 |
| Central | Anhui | 119.78 | 19,354.21 | 32,874.98 |
| | Jiangxi | 778.18 | 11,144.53 | 47,231.87 |
| | Henan | 48.78 | 334.20 | 804.72 |
| | Hubei | 355.88 | 6043.55 | 7185.26 |
| | Hunan | 569.60 | 12,428.53 | 50,716.87 |
| | Mean | 374.50 | 9860.95 | 27,762.74 |
| West | Shanxi | 39.46 | 1246.20 | 1296.51 |
| | Sichuan | 398.40 | 31,065.84 | 65,616.13 |
| | Chongqing | 49.05 | 1181.45 | 8574.45 |
| | Yunnan | 37.10 | 8031.11 | 16,979.64 |
| | Guizhou | 212.05 | 3603.12 | 3545.31 |
| | Guangxi | 122.42 | 21,994.05 | 51,965.99 |
| | Mean | 143.13 | 11,186.92 | 24,663.08 |
| Total | - | 3929.82 | 218,743.23 | 502,446.85 |

Data source: *China Forestry Statistical Yearbook*.

To dissect the growth and change in the BHTI output value, we used ArcGIS to draw a map of the growth rate change. Figure 3 illustrates the growth rates from 2000 to 2010, with darker colors indicating higher growth rates. Overall, the average growth rate was 93.01% in the eastern region, 43.31% in the central region, and 60.69% in the western region. Specifically, Zhejiang Province in the eastern region experienced a growth rate of 83.31%, increasing from USD 3.99 million in 2000 to USD 336.59 million in 2010. In the western region, Yunnan Province demonstrated a remarkable growth rate of 215.46%, from USD 371,000 to USD 803 million. In contrast, the central region, represented by Henan Province, showed a lower growth rate of only 5.85%. These results indicate that during the period 2000–2010, BHTI in the eastern region exhibited robust growth, while the industry in the western region was still in its infancy, and the growth in the central region even slowed down. The major driving force of the BHTI lies in the demand from small and medium-

sized bamboo processing enterprises in the central and eastern regions and the abundant bamboo resources in the western region.

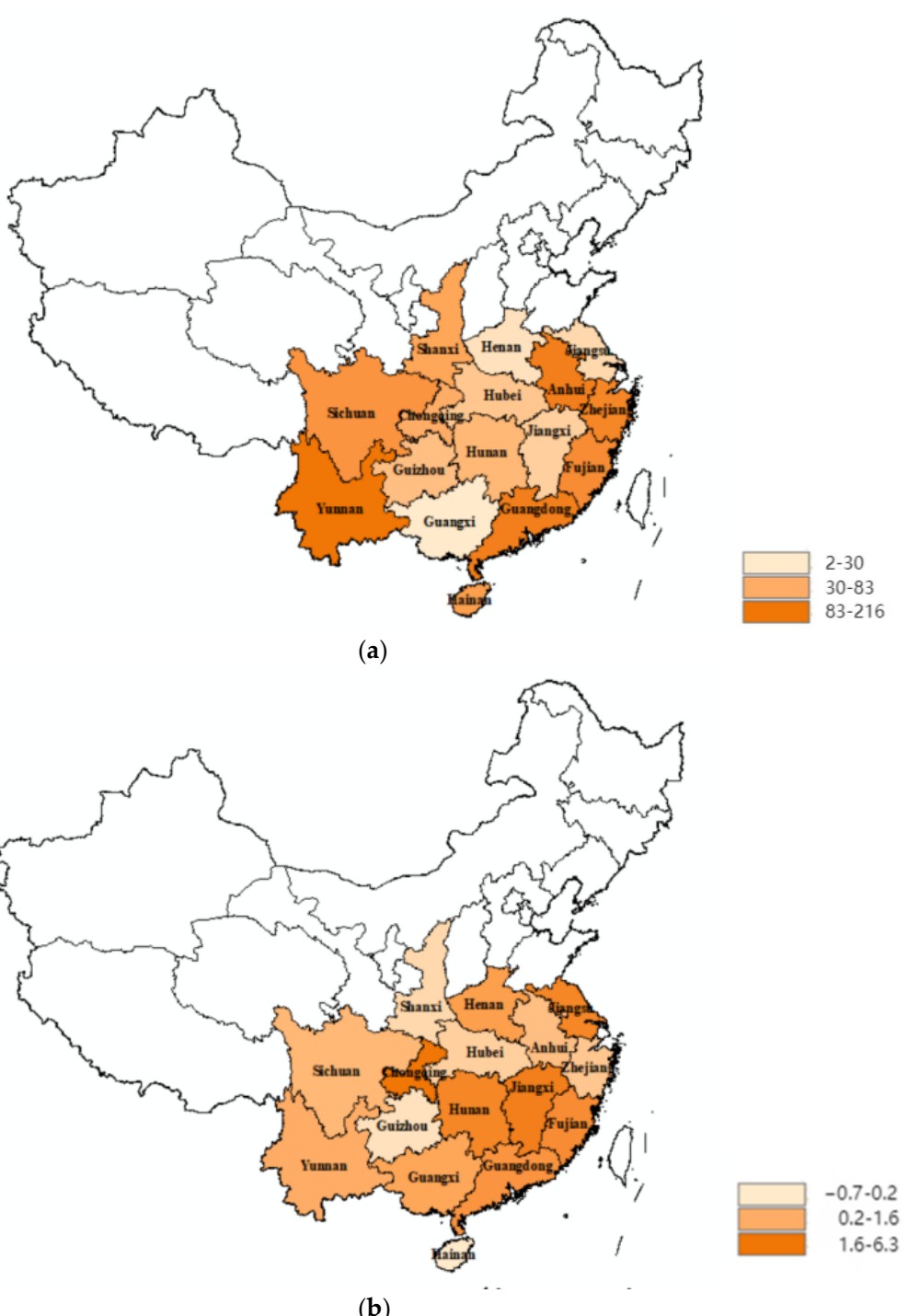

(**a**)

(**b**)

**Figure 3.** Growth rate of bamboo harvesting and transportation output value in 16 provinces during 2000–2010 and 2010–2020 (%). (**a**) 2000–2010. (**b**) 2010–2020.

As shown in Figure 3, the growth rate of BHTI output value declined in several eastern provinces, including Zhejiang and Hainan. Specifically, the growth rate in Zhejiang decreased from 83.31% to 0.2%, while that in Hainan shifted from 60.4% to negative. It can be attributed to the heightened environmental regulations in the eastern region. This, in turn, resulted in reduced demand for the primary production and transportation of bamboo forests, eventually lowering the output value of local BHTI. In contrast, the growth

rate in the central and western regions increased, exemplified by Jiangxi, Hunan, and Chongqing. This trend suggests that strong environmental regulation in the bamboo industry may compel bamboo-related activities to shift from the eastern to the central and western regions.

Table 4 illustrates the variations in MER intensity across the 16 provinces in 2000, 2010, and 2020, and the average pollutant discharge fee in the eastern region surpassed that of the central and western regions. To be specific, the average fee in the eastern region exhibited a substantial increase of 129% from 2000 to 2010, then a modest 13% rise from 2010 to 2020, suggesting a sluggish development of the bamboo processing industry in the face of stringent MER policies. In the central region, the average pollutant discharge fee also surged from 2000 to 2010, signifying substantial MER intensity, but decreased from 2010 to 2020, implying a relaxation of MER policies. The pollutant discharge fee in the less economically developed western region showed an upward trend, though the average fee was lower than that in the eastern and central regions. The modest increase from 2010 to 2020 suggested growth potential in bamboo harvesting, transportation, and processing in regions with lower MER intensity—the central and western regions. Furthermore, the bamboo industry also showed a tendency to transfer from the eastern to the western region.

**Table 4.** Variations in the MER intensity in the 16 provinces in 2000, 2010, and 2020.

| Region | Province/City | Pollutant Discharge Fee of Provinces (Unit: USD Ten Thousand) | | |
|---|---|---|---|---|
| | | **2000** | **2010** | **2020** |
| Eastern | Zhejiang | 7156.69 | 13,347.17 | 3906.17 |
| | Jiangsu | 7115.01 | 27,624.99 | 48,973.49 |
| | Fujian | 2793.19 | 4919.50 | 4124.20 |
| | Guangdong | 8365.69 | 12,284.65 | 7905.86 |
| | Hainan | 194.56 | 490.55 | 1151.05 |
| | Mean | 5125.03 | 11,733.37 | 13,212.09 |
| Central | Anhui | 1834.34 | 7184.49 | 4481.48 |
| | Jiangxi | 1403.55 | 6819.85 | 4883.79 |
| | Henan | 3932.71 | 11,780.06 | 12,561.32 |
| | Hubei | 2793.19 | 5141.98 | 8116.95 |
| | Hunan | 3112.82 | 7375.57 | 5794.15 |
| | Mean | 2615.32 | 7660.45 | 7167.54 |
| Western | Shanxi | 2153.96 | 6450.89 | 5747.73 |
| | Sichuan | 2195.65 | 7659.33 | 8138.07 |
| | Chongqing | 1208.99 | 4992.18 | 3762.76 |
| | Yunnan | 1459.13 | 3916.03 | 8174.34 |
| | Guizhou | 1070.03 | 6237.17 | 8259.25 |
| | Guangxi | 3446.33 | 5839.73 | 5816.10 |
| | Mean | 1922.30 | 5848.84 | 6649.23 |
| Total | - | 50,232.96 | 132,056.53 | 141,788.53 |

Source of data: *China Environmental Statistics Yearbook*.

### 4.2. Results and Discussion

This study used Stata15.0 to conduct fixed effects model regression analysis. The results are shown in Table 5. Before the regression analysis, we conducted an inter-group heteroscedasticity test and an intra-group autocorrelation test on the original panel data. The *p*-values were less than 0.1, so there was no heteroscedasticity and autocorrelation. Therefore, the T value reported in the table is correct and can be used to judge whether the variable is significant.

**Table 5.** Regression results of influence of MER intensity on the output value of BHTI.

| Variables | Dependent Variable: BHTI Output Value | | |
| --- | --- | --- | --- |
| | **Total** | **Before 2013** | **After 2013** |
| MER | −1.062 *** | −1.533 *** | −0.336 * |
| | (0.271) | (0.282) | (0.199) |
| *pop* | −873.1 *** | −1253 *** | −2447 ** |
| | (314.3) | (328.5) | (1037) |
| *inc* | 15.86 *** | 39.55 *** | 12.05 * |
| | (5.316) | (6.034) | (6.541) |
| *stru* | $-8.064 \times 10^8$ | $-1.452 \times 10^9$ | $1.106 \times 10^{10}$ |
| | $(3.696 \times 10^9)$ | $(2.337 \times 10^9)$ | $(1.086 \times 10^{10})$ |
| *tem* | −7973 | −6634 | 18,495 |
| | (21,018) | (12,408) | (30,406) |
| *pre* | 6.524 | −42.99 * | −14.50 |
| | (47.27) | (25.28) | (45.36) |
| Regional fixed effect | Controlled | Controlled | Controlled |
| Time-fixed effect | Controlled | Controlled | Controlled |
| Constant | 462,790 | 562,981 ** | 547,534 |
| | (351,758) | (240,141) | (493,678) |
| Observations | 336 | 208 | 128 |
| R-squared | 0.734 | 0.835 | 0.929 |

Note: ***, **, and * represent the significance levels of 1%, 5%, and 10%, respectively. The standard error is in brackets.

In detail, the results indicate a significant negative impact of MER intensity on the BHTI output value at a 1% significance level, proving Hypothesis I. In other words, heightened MER intensity diminished the output value and hindered the progress of the BHTI. Pollutant discharge fee, a major MER tool, proved to be effective in pollution reduction, yet an excessive discharge fee may render enterprises unable to afford the costs and have to close down. Consequently, pollutant discharge fee can reduce the primary production of bamboo harvesting and transportation. This result is consistent with some previous studies. Environmental regulations significantly inhibit the development of regional industries [52], and stricter environmental regulations will strengthen the negative effects on enterprises [47], mainly because pollution emissions will increase the cost of environmental protection, leading to difficulties in enterprises. However, some studies, different from this result, found that MER has a regulatory effect [53], which can show positive development through industrial green technology innovation [54], which may be due to the greater economic benefits brought by technological upgrading.

As the regression results of control variables, population density demonstrated a significant negative impact on the BHTI output value at a 1% significance level. The higher the population density, the lower the output value. The reason may be that as the population density rose, the demand for land increased, leading to more fierce competition for limited resources and disputes over land use and access. Furthermore, higher population density often drives urbanization and places pressure on available land for housing, infrastructure, and commercial purposes. This is the same as a previous study, which concluded that population density dominates the green space and affects local economic development [55]. Per capita income, on the other hand, demonstrated a significant positive impact on the BHTI output value at a 1% significance level. The higher the income from bamboo production, the fewer liquidity constraints for bamboo farmers and the more willing they are to invest, thus lifting up the BHTI output value. This is the same as a previous study, which has shown that such agricultural subsidies increased the per capita income of farmers, which helped farmers to operate and promote the development of their industries [56]. Next, average annual temperature and precipitation showed no significant impact on the BHTI output value. Other factors like the bamboo growth environment, as well as economic and social factors, might play a role. Additionally, industrial structure exhibited no significant impact.

It should be noted that the Chinese government called for the construction of ecological civilization and issued 14 specifications to combat environmental pollution since 2013. Enterprises and the public have more incentives for environmental protection by government regulation. Therefore, this study conducted a regression analysis of the influence of MER intensity on the BHTI output value before and after 2013. The results are shown in Table 5.

Compared with before 2013, MER had a less significant negative impact on BHTI after 2013, which proved that BHTI gradually adapted to the MER due to the Chinese government being more focused on reducing environmental pollution since 2013. Annual average precipitation exerted a significant negative influence at a 10% significance level before 2013. To put it differently, unfavorable conditions for bamboo growth are associated with a higher level of precipitation. Additionally, increased precipitation made it more challenging to discharge pollutants, leading to higher pollutant discharge fee, which, in turn, reduced the demand for bamboo harvesting and transportation. Similarly, after 2013, the impact of population density on the BHTI output value was significantly negative at a 5% significance level, while the influence of per capita income was significantly positive. The industrial structure showed no significant impact. This is basically consistent with the research hypothesis.

The robustness test has been given to prove the reliability of the regression above. The pollutant discharge fee of the year is employed as the proxy variable for MER intensity to analyze its impact on BHTI output value. Considering the potential lag effect of MER, to further validate the robustness of the empirical results, this study employed a one-period lag to measure key explanatory variables. Specifically, we investigated the influence of the current period's pollutant discharge fee (*L. fee*) on the BHTI output value of the next period. The regression results are shown in Table 6, indicating that the robustness test results are generally consistent with the above baseline empirical results, confirming the model's robustness.

**Table 6.** Robustness test of influence of MER intensity on BHTI output value.

| Variables | Dependent Variable: BHTI Output Value |
|---|---|
| *L. fee* | −0.662 *** |
| | (0.230) |
| *pop* | −69.08 |
| | (280.6) |
| *inc* | 5.712 *** |
| | (1.101) |
| *stru* | $-5.343 \times 10^9$ * |
| | $(2.732 \times 10^9)$ |
| *tem* | −3573 |
| | (14,647) |
| *pre* | −3.585 |
| | (31.37) |
| Regional fixed effect | Controlled |
| Time-fixed effect | Controlled |
| Constant | 192,732 |
| | (276,235) |
| Observations | 320 |
| R-squared | 0.911 |

Note: *** and * represent the significance levels of 1% and 10%, respectively. The standard error is in brackets.

The analysis above proved that MER intensity yields an inhibitory effect on the BHTI output value. Given the difference in pollutant discharge fee and environment protection policies across various regions in China, the influence of MER intensity varies. By analyzing the spatiotemporal evolution of such influence, we found a prominent heterogeneity among eastern, central, and western regions. Therefore, we categorized the 16 provinces into

eastern, central, and western regions, and carried out a quantitative analysis to explore the regional heterogeneity in the correlation between MER intensity and BHTI output value.

Regression results in Table 7 have been given to find the different impacts of BHTI from MER in regions with different levels of economic development. It can be observed that MER has a more significant negative impact on BHTI in eastern provinces compared to central and western provinces. This suggests that the impact of MER intensity on the BHTI output value was significantly negative at a 1% level in the eastern region but insignificant in the central and western regions. With a higher level of economic development, bamboo enterprises in the eastern region of China usually faced higher scrutiny and compliance costs imposed by stringent environmental regulations, which would force bamboo processing enterprises to transfer to the central and western regions. Jiangxi and Hubei in the central region and Sichuan in the western region are rich sources of bamboo resources [13] and have implemented preferential policies to boost the bamboo industry in recent years. Meanwhile, Zhejiang made more efforts to reduce pollution and improve the environment but ended up forcing local bamboo enterprises to transfer to the central and western regions. On the other hand, the output value in central and western regions showed an upward trend, verifying Hypothesis 2. In the past, regional economic development was brought about by the large consumption of resources and environmental pollution. The strengthening of environmental regulation in these areas will lead to its transfer to areas with weak environmental regulation and a relatively backward economy [52].

**Table 7.** Regression results of the influence of MER intensity on BHTI output value in the eastern, central and western regions.

| Variables | Dependent Variable: BHTI Output Value | | |
| --- | --- | --- | --- |
| | East | Central | West |
| MER | −1.579 *** | −0.637 | −0.655 |
| | (0.324) | (0.624) | (1.045) |
| *pop* | −3124 *** | −333.6 | −2266 ** |
| | (703.4) | (699.0) | (1082) |
| *inc* | −0.822 | 38.86 *** | 15.72 |
| | (5.615) | (8.419) | (24.62) |
| *stru* | $4.915 \times 10^{10}$ *** | $-1.190 \times 10^{10}$ *** | $-1.552 \times 10^{10}$ ** |
| | $(8.701 \times 10^{9})$ | $(3.575 \times 10^{9})$ | $(5.895 \times 10^{9})$ |
| *tem* | 74,187 * | 19,884 | 15,854 |
| | (43,308) | (36,923) | (21,361) |
| *pre* | 3.222 | 39.48 | 122.1 * |
| | (73.96) | (62.71) | (64.00) |
| Regional effect | Control | Control | Control |
| Time effect | Control | Control | Control |
| Constant | $-1.169 \times 10^{6}$ | −110,964 | 378,363 |
| | (883,051) | (697,193) | (420,324) |
| Observations | 126 | 105 | 105 |
| R-squared | 0.830 | 0.890 | 0.718 |

Note: ***, **, and * represent the significance levels of 1%, 5%, and 10%, respectively. The standard error is in brackets.

## 5. Conclusions and Recommendations

MER, as a regulatory approach, is effective in encouraging enterprises to adopt environmentally friendly practices of pollution reduction, but it may also yield a negative effect due to increased operational costs of pollution control. This study, based on the externality theory and the Pollution Haven Hypothesis, ran a spatiotemporal analysis of the BHTI output value of 16 provinces in China from 2000 to 2020. In addition, the econometric model was used to analyze the spatiotemporal impact of MER on the output value.

First, the MER intensity and the BHTI output value in the eastern region were generally larger than those in the central and western regions. The eastern region experienced

a gradual increase in output value and stricter intensity of environmental regulation, whereas the central and western regions saw a substantial increase in output value but relaxed intensity of regulation. Second, MER exerted a significantly negative impact on the BHTI output value. Via economic mechanisms such as pollutant discharge fee, MER compels enterprises to adopt environmentally friendly production technologies and management practices. This necessitates technological upgrades and innovation, entailing increased capital and technology investments. Consequently, production costs rise, and some enterprises have to close, causing the BHTI output value to drop. Third, when examining the regression results of the three regions, the impact of MER on the BHTI output value was statistically significant in the eastern region but insignificant in the central and western regions. The environmental regulation intensity in the central and western regions remained lenient, driving the BHTI to expand further. However, the well-established bamboo industry in the eastern region, coupled with tightened environmental regulations, resulted in stricter environmental policies. As a result, the BHTI gradually shifted from the eastern region to the central and western regions.

China is facing a transition of development for environmental protection; the Chinese government is paying more attention to environmental regulation of industries to improve green development. Although many livelihoods of rural households in developing countries still depend on bamboo or other wood manufacturers, they need to face the balance of environmental protection and industry development in the future. Based on the above conclusions, this study proposes the following policy recommendations for the bamboo industry in developing countries. First, it is imperative to make long-term planning for the transformation and development of the bamboo industry. The government should increase policy support and provide scientific management and macro guidance to enterprises and farmers related to the bamboo industry. Second, MER should be paid more attention to so the governments in environmental regulation policies could improve efficiency. With stricter environmental regulations in the future, it is necessary to build an industry transfer plan for bamboo manufacturers; bamboo manufacturers in the developed region can be gradually transferred to the developing region to reduce the cost of pollutant emission. Third, it should also strengthen the infrastructure construction of bamboo forest planting, select and breed superior varieties, and introduce relevant talents, so as to effectively solve various environmental problems existing in the processing industry.

In conclusion, it is important to acknowledge the limitations of this study. First, as for the measurement index of MER, this paper only uses the pollutant discharge fee, which is relatively simple, and other variables can also be used as multiple indicators to measure. Second, the MER selected in this paper is only one kind of environmental regulation, and the mandatory environmental regulation of the government can be considered. Finally, the MER studied in this paper has a certain lag effect on the output value of bamboo forest harvesting and transportation. At present, the observation value used is only a stage of panel data, which also needs continuous observation and follow-up. In future studies, with more data and a broader range of research subjects, additional indicators, such as pollutant discharge permit or environmental subsidy, will be incorporated. This expansion aims to provide a more comprehensive assessment of MER intensity, thereby enhancing the overall scope of measurement in subsequent studies.

**Author Contributions:** Conceptualization, Z.Z. and C.C.; methodology, Z.Z.; software, T.G.; validation, T.G.; formal analysis, Z.Z., T.G. and C.C.; resources, Z.Z., T.G. and C.C.; data curation, T.G. and C.C.; writing—original draft preparation, T.G. and C.C.; writing—review and editing, T.G. and Z.Z.; supervision, Z.Z. All authors have read and agreed to the published version of the manuscript.

**Funding:** This research was funded by the Zhejiang Social Science Fund, grant number 23QNYC13ZD.

**Data Availability Statement:** All data generated or analyzed during this study are included in the article.

**Acknowledgments:** We are very grateful to all the reviewers, institutions, and researchers for their help and advice in relation to our work.

**Conflicts of Interest:** The authors declare no conflicts of interest.

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
