# Peer review of "How Market-Oriented Environmental Regulation Impacts the Bamboo Industry in China"

_forests, doi:10.3390/f15030457_

Round 1

Reviewer 1 Report

Comments and Suggestions for Authors

1.      Shorten the title, and it should be catchy to attract the readers' attention.  

2.      Describe the type of data collected and analysed in the abstract and conclude with the contribution/ originality of this study.

3.      To promote ecological civilisation construction, environmental policies, and regulations such as industry waste water treatment and pollutant discharge fee have been imposed on pollution intensive enterprises to address environmental challenges and pro mote sustainable development (pg 2, lines 60-63). This sentence can be improved in a few ways, such as (1) clarity and English. (2) Split into two sentences, and (3) it is essential to justify the type of fee to address the environmental challenges.

4.      From many alternatives in the literature, why did authors choose Market-oriented environmental regulation compared to existing solutions?

5.      However, few studies have investigated the influence of MER on a regional transfer within the bamboo industry. Discuss the results and gaps of past studies. Which gaps do your study need to focus on?

6.      Include the research questions and highlight the significance of this study in theory and practice. The most important highlight is the novelty and uniqueness.

7.      Include the theory to conceptualise how the authors identify the variables and discuss the hypothesis development for each.

8.      The conceptual framework needs better justification and demands supporting reference.

9.      Discuss the validity and reliability of individual and time effects assessment criteria.

10.  It is not necessary to include the null hypothesis. It is not stated in the hypotheses. Delete it.

11.  What are the reasons data is not available from 4 provinces? Does the collected data represent the industry?

12.  Explain what type of data was collected. What is the procedure of data collection? What is unit analysis?

13.  It is good when the profile of samples is included in the analysis.

14.  Strengthen discussion of theoretical implications and necessary suggestions to improve current practices in the bamboo industry. Based on the findings, what the government should consider? 

Comments on the Quality of English Language

To promote ecological civilisation construction, environmental policies, and regulations such as industry waste water treatment and pollutant discharge fee have been imposed on pollution intensive enterprises to address environmental challenges and pro mote sustainable development (pg 2, lines 60-63). This sentence can be improved in a few ways, such as (1) clarity and English. (2) Split into two sentences, and (3) it is essential to justify the type of fee to address the environmental challenges. Many more can be improved.  

Author Response

Thank you very much for your revision suggestions. Regarding the suggestions and issues you mentioned, I have made the necessary changes in the document and highlighted them with a yellow background. I have also made annotations on the side. If you have any questions, please feel free to provide further suggestions.

Reviewer 2 Report

Comments and Suggestions for Authors

This paper investigates the effect of environmental regualtions on tharvesting, trasnport and processing of bamboo in some provinces.  Unfortunately, the manuscript does not describe neither the sectors nor the environmental regulatons. Additionally, to consider spatial effects i  strongly encourage the authors to consider the use of spatial econometrics. 

I find the effects of MER  on output value interesting but the manuscript should explain more in detail the MER.   How is the MER exactly defined for the different provinces and each industry: harvesting, transportation and  processing. Are there environmental regulations ?

The description of the industries is poor and basicaly only inscludes references. Would it be possible to describe  in the introduction the harvesting and its environemntal damages, trasnportation and its environmental damages, processing and its enviroanmental damages. The same in the resutls and coclusion sections. Are harvesting more affected or the other industries (transportation o processing).

Could you please specified the measure you use as output  value:  is it revenue, net revenue, value added? 

Author Response

Thank you very much for your question. Regarding the suggestions and issues you mentioned, I have made modifications in the document and highlighted them with a green background. I have also made annotations on the side. What I also want to explain here is that the harvesting and transportation mentioned in the article is counted as an industry in China's statistical yearbook, so the research object is the bamboo harvesting and transportation industry, and the research question is the impact of market-oriented environmental regulations(MER) on it. If you have any questions, please feel free to provide suggestions again.

Reviewer 3 Report

Comments and Suggestions for Authors

This manuscript provides valuable insights into the impact of environmental regulations on the bamboo industry. It includes a conceptual framework, methods, results, discussions, and conclusions, which are essential components of a technical research article. However, some sections need to be better structured by revising the manuscript for clarity and completeness. In particular, the manuscript is missing a large number of references. The manuscript needs a major revision. It is unlikely to be accepted in this form. To enhance the manuscript, I would like to suggest the following:

1) The manuscript appears to be written in a disorganized manner, with issues such as concatenated words, spacing errors, and most importantly, a significant number of missing references (W et al. 2023, Bobby et al., 2023, Lvwen, 2021, and many others). Similar problems are present in the references section, including citations not mentioned in the text (Zhouhao S, Dong R, Chengyou L, et al. Agricultural subsidies on common prosperity…, Zsarnóczai S J, Zéman Z. Output value and productivity of the agricultural industry in Central-East…, etc.), incorrect formatting, and alphabetical disorder.

2) The novelty of the manuscript should be highlighted more effectively.

3) Are the data for the selected variables between 2000 and 2020 annual data? Have they met the necessary assumptions to conduct the tests? e.g. normality distribution for the f-test?

4) In general, the coefficient of determination (R2) is calculated to assess how strong the linear relationship is between two variables in regression results. The coefficient of determination is used to explain the relationship between an independent and dependent variable. Include R2 values that indicate how much your selected independent variables explain the dependent variables.

5) What is the reason for providing Table-1 and Table-3 as appendix?

6) The manuscript, in its current form, appears somewhat simplistic. Future predictions can be made using the established regression models.

Author Response

Thank you very much for your suggestions. Regarding the suggestions and issues you mentioned, I have made modifications in the document and highlighted them with a blue background. I have also made annotations on the side. In addition, I have made significant revisions to the references to ensure that each reference mentioned in the article corresponds one-to-one in the references. If you have any questions, please feel free to provide suggestions again.

Round 2

Reviewer 1 Report

Comments and Suggestions for Authors

It looks better now. 

Comments on the Quality of English Language

It looks better now. 

Author Response

Thank you for your suggestion.

Reviewer 2 Report

Comments and Suggestions for Authors

I don´t have any new comment.

Author Response

I have explained your question in more detail in cover letter. If you have more questions, please feel free to continue to ask.

Reviewer 3 Report

Comments and Suggestions for Authors

The manuscript has improved a lot and many of the points have been met satisfactorily. Still, there are missing issues that I believe need to be resolved or clarified for the article to pass.

1- Did your data set meet the assumptions necessary to perform statistical tests? Ex. homogeneity, normality distribution?

2- I still cannot see the relationship between the independent variables and the dependent variable. Calculate R2 (determination coefficient)

Author Response

Thank you for your suggestion.
